# Small Dense LDL: Scientific Background, Clinical Relevance, and Recent Evidence Still a Risk Even with ‘Normal’ LDL-C Levels

**DOI:** 10.3390/biomedicines10040829

**Published:** 2022-04-01

**Authors:** Harold Superko, Brenda Garrett

**Affiliations:** Cholesterol, Genetics, and Heart Disease Institute, Carmel, CA 93923, USA; superbrenda@mac.com

**Keywords:** low density lipoprotein, cholesterol, cardiovascular disease, residual risk, small LDL, lipoprotein heterogeneity, sdLDL, coronary artery disease

## Abstract

Residual cardiovascular disease event risk, following statin use and low-density lipoprotein cholesterol (LDL-C) reduction, remains an important and common medical conundrum. Identifying patients with significant residual risk, despite statin drug use, is an unmet clinical need. One pathophysiologic disorder that contributes to residual risk is abnormal distribution in lipoprotein size and density, which is referred to as lipoprotein heterogeneity. Differences in low density lipoprotein (LDL) composition and size have been linked to coronary heart disease (CHD) risk and arteriographic disease progression. The clinical relevance has been investigated in numerous trials since the 1950s. Despite this long history, controversy remains regarding the clinical utility of LDL heterogeneity measurement. Recent clinical trial evidence reinforces the relevance of LDL heterogeneity measurement and the impact on CHD risk prediction and outcomes. The determination of LDL subclass distribution improves CHD risk prediction and guides appropriate treatment.

## 1. Introduction

Despite significant, medication induced reduction in low density lipoprotein-cholesterol (LDL-C), a large reservoir of cardiovascular disease risk remains. The often quoted 25% relative risk reduction, attributed to LDL-C reduction, is actually only a 3.4% absolute risk reduction (Figure 1) [1].

For example, in the JUPITER trial, rosuvastatin achieved a 50% reduction in LDL-C, and there were 251 primary endpoints in the placebo group, yet 142 subjects experienced a primary end point in the treatment group despite a 50% reduction in LDL-C [2]. In the Fourier investigation, PCSK9 inhibition, in addition to statin therapy, reduced LDL-C to a mean of 30 mg/dL compared to 92 mg/dL in the statin only group [3]. There were 1563 primary endpoints in the statin control group, yet 1344 subjects in the PCSK9+statin group also experienced a primary endpoint. This represents a primary endpoint event rate of 9.8% in the PCSK9 group compared to 11.3% in the control group. Despite very successful LDL-C reduction with the PCSK9 inhibitor, the absolute risk reduction of 1.5% reflects an unmet clinical need to identify factors contributing to CHD risk other than standard blood lipid measurements. Several metabolic disorders have been proposed that may account for a substantial portion of the residual risk. One such disorder is an abundance of small, dense LDL particles which has been investigated since the 1950s and for which recent clinical trial evidence has strengthened the clinical relevance as a nontraditional marker of CHD risk.

The purpose of this review is to summarize historic and recent clinical trial evidence that relates to the importance of small, dense, LDL (sdLDL) in the prediction of cardiovascular disease risk, treatment response, and clinical outcomes.

## 2. Clinical Relevance of LDL Heterogeneity

Cardiovascular disease risk is associated with elevations in LDL-C. However, many myocardial infarction patients have a LDL-C that would be considered normal in a primary prevention population. According to the American Heart Association’s “Get With The” program, approximately 75% of patients admitted to hospital with a coronary heart disease (CHD) event exhibited a relatively normal LDL-C less than 130 mg/dL (3.36 mmol/L) and 23% had a LDL-C less than 70 mg/dL (1.61 mmol/L) [4]. Thus, many patients remain at risk for a CHD event even when LDL-C is in an acceptable range and a new strategy to prevent myocardial infarctions is required [1]. Lipoproteins include a heterogeneous mixture of particles that vary by size, density, buoyancy, and composition. The role of small, dense, LDL (sdLDL) in the pathology of coronary heart disease (CHD) has previously been described [5].

## 3. History of Lipoprotein Heterogeneity and Laboratory Methods

Dr. John Gofman and colleagues first described differences between subclasses of LDL and HDL and the relationship to CHD risk in the Framingham and Livermore studies that were established between 1955 and 1957 [6,7,8]. Over the next 65 years, multiple investigations were undertaken by a variety of investigators. Originally, differences in lipoproteins within the traditional VLDL, IDL, LDL, and HDL density regions were explored using analytic ultracentrifugation (ANUC). Subsequently, a variety of other laboratory techniques have been utilized to assess lipoprotein heterogeneity including polyacrylamide gradient gel electrophoresis (PAGGE), density gradient ultracentrifugation (DGU), precipitation assays, surfactant phospholipase assay (SPA), nuclear magnetic resonance (NMR), and ion mobility (IM). Each of these laboratory methods determines different characteristics of lipoprotein heterogeneity. ANUC determines particle density as measured by Svedberg floatation intervals represented on a Schlerian curve, and while very precise, its cost and low sample through-put limit its clinical application [9]. PAGGE is an electrophoresis method that separates particles based on a gradation in pour size along a gradient [10]. This method requires a specific gel gradient, standards, staining, and scanning to determine percent distribution and peak particle diameter, but it is not quantitative. DGU is a standard research laboratory method that separates particles based on density and then the determination of cholesterol content within regions of interest [11]. sdLDL-C is a method that relies on the use of surfactants and enzymes that selectively react with certain groups of lipoproteins with results reported in mg/dL [12]. NMR is a method that relies on a library of reference spectra of lipoprotein subclasses incorporated into a linear least-squares fitting computer program which works backward from the shape of the composite plasma methyl signal to compute the subclass signal intensities [13]. While each method determines unique characteristics of lipoprotein subclasses, a comparison study of four methods in the HDL-Atherosclerosis Treatment Study (HATS), reported that the presence of excessive amounts of small, dense LDL, by any of the four methods, was associated with increased atherosclerosis risk [14].

### Indirect Approximation of LDL Heterogeneity

Ratios of easily obtainable standard laboratory measurements have been proposed as a means to estimate small LDL, including HDL-C, fasting triglycerides, triglycerides/HDL-C, and LDL-C/Apolipoprotein-B. One standard measurement, fasting triglyceride, has revealed a significant statistical correlation to LDL size, which can weaken but not eliminate the independent relationship between LDL size and cardiovascular risk [15]. This correlation may be clinically useful in predicting elevated sdLDL levels when fasting triglycerides are greater than 250 mg/dL, and predicting the lack of an abundance of sdLDL when fasting triglycerides are less than 70 mg/dL. The recent ACC consensus document on hypertriglyceridemia suggests that fasting triglyceride values consistently >150 mg/dL be the cut point at which to take clinical action [16]. However, an analysis of 5366 CHD patients and fasting triglycerides and LDL size revealed that below a fasting triglyceride level of 70 mg/dL, only 4.2% of subjects expressed small LDL, but for the group with fasting triglycerides <150 mg/dL, 22.7% also expressed small LDL, which reflects increased CHD risk. Conversely, in subjects with fasting triglycerides between 150–250 mg/dL, 79.1% expressed small LDL and in subjects with fasting triglycerides >250 mg/dL, and 100% revealed small LDL (Figure 2). 

Thus, assuming fasting triglycerides < 150 mg/dL reflects the absence of small LDL can be an error in approximately 27% of patients. Most recently, a new small LDL prediction equation, based on standard lipid panel results, has been proposed to enhance CHD risk estimation [17].

## 4. Evidence for LDL Heterogeneity Association with CHD Risk

Identification of a group of individuals with an abundance of small, dense LDL identifies individuals that carry a significantly increased CHD risk [18]. An abundance of small, dense LDL particles is a common finding in the CHD patient population, with 30–40% of CHD patients expressing an abundance of small LDL. In the Quebec Cardiovascular Study, statistical adjustment for LDL-C, triglycerides, HDL-C, and apoB had virtually no impact on the relationship of small LDL and CHD risk [19]. However, the presence of other risk factors, such as elevated Apo B, magnified the risk associated with small, dense LDL. The increased CHD risk associated with an abundance of small, dense LDL has been reported, consistently and reproducibly, in the Boston Area Heart study, the Stanford Five City project, the Harvard Physicians Health Study, the Quebec Cardiovascular study, and the Women’s Health Study, among others (Table 1) [18,20,21,22,23]. Relatively recent clinical trial reports have confirmed the atherogenicity of small LDL in the Malmo Heart Study in 2009, as well as the Atherosclerosis Risk in Communities (ARIC) and the Multi-Ethnic Study of Atherosclerosis (MESA), both in 2014 [24,25,26]. Indeed, nontraditional markers of cardiovascular disease risk may improve the 2013 American College of Cardiology/American Heart Association guidelines [27].

### 4.1. Small, Dense LDL-C Quantitative Level and CHD Risk

Notably, in ARIC small, dense LDL levels greater than 50 mg/dL (1.29 mmol/L), predicted risk, even in individuals with a LDL-C < 100 mg/dL (2.58 mmol/L), which normally would be considered to reflect low CHD risk [25]. The risk is graded and ARIC results revealed that stepwise quartile increases in sdLDL-C, in subjects with a mean LDL-C of 122 mg/dL, was associated with increased CHD risk while the quartiles of large LDL revealed no stepwise difference in CHD risk (Figure 3). 

In ARIC an abundance of sdLDL-C predicted CHD events, even in the group with LDLL-C < 100 mg/dL who were initially felt to be at low CHD risk. The MESA study results were similar and reported a significant increased CVD risk when sdLDL-C was >46 mg/dL [26]. The level at which sdLDL-C contributes to CHD risk may be lower than 50 mg/dL. In a study, conducted in Japan, LDL-C values above or below the median of 100 mg/dL did not predict CHD risk but a sdLDL-C above the median of 35 mg/dL revealed a significantly increased risk in stable CHD patients [41]. 

### 4.2. Independence of Small, Dense LDL-C as a Risk Predictor

In patients with established CHD, elevations in the smallest LDL region have been reported to be the single best predictor of increased coronary artery stenosis [34,42]. As far back as 1992, the Saint Thomas Atherosclerosis Regression Study (STARS) reported that after cholestyramine treatment, a change in dense LDL (small LDL) was the best predictor of CHD outcome defined by coronary arteriography [29]. The HATS trial reported that when results were adjusted for risk factors, the odds for primary clinical CHD endpoints were significantly greater in subjects with higher on-study small LDL levels both before (*p* = 0.01) and after (*p* = 0.03) adjustment for the treatment group and the standard lipid values. In patients undergoing coronary artery bypass surgery, progression of underlying disease is a concern. The Emory Angioplasty and Surgery Trial (EAST) reported that in multivariate analysis, native coronary disease progression was independently correlated with small LDL particles [35].

### 4.3. Statin Therapy and sdLDL-C CHD Risk

Elevated sdLDL-C remains a risk predictor even in patients on statin therapy. In patients with established CHD, Sakai and colleagues have shown that a significant association of sdLDL-C and CV events was observed in statin users (HR 1.252, 95% CI 1.017–1.540), diabetes patients (HR 1.219, 95% CI 1.018–1.460), patients without diabetes (HR 1.257, 95% CI 1.019–1.551) and patients with hypertriglyceridemia (HR 1. 376, 95% CI 1.070–1.770) [40]. They concluded that sdLDL-C was the most effective predictor of residual risk of future cardiovascular events in stable coronary artery disease patients using statins.

The Justification for the Use of Statins in Prevention: An Intervention Trial Evaluating Rosuvastatin (JUPITER) study confirmed the CHD risk associated with small LDL. Even in patients treated with rosuvastatin and with an average LDL-C of 54 mg/dL, a significant increase in risk for CHD and all-cause death was associated with small LDL [37].

### 4.4. Small Dense LDL, Not Always a Significant Predictor of CVD Events

Small dense LDL-C is not a significant predictor of CVD events in all patient sub-groups. In patients with low HDL-C and well controlled LDL-C, sdLDL-C as a biomarker did not predict future CVD events in a secondary analysis of the AIM-HIGH trial [38]. In this investigation, the LDL-C in the group with no cardiovascular events, compared to the group with cardiovascular events, was 71.1 mg/dL and 71.8 mg/dL (*p* = 0.29), and the sdLDL-C was 32.7 mg/dL and 32.9 mg/dL respectively. This analysis provides support to the concept that below a yet to be defined threshold, the absolute amount of sdLDL-C loses its significance as a predictor of CV events.

### 4.5. Non-Invasive Imaging and Small, Dense LDL

Noninvasive imaging tools can identify patients with subclinical atherosclerosis. Carotid artery intima-media thickness measurement has revealed a significant relationship between sdLDL-C and vessel wall thickness [43]. The presence of coronary artery calcium, in patients at intermediate cardiovascular risk, can provide additional predictive information to that of conventional risk factors [44]. In the healthy women study, small LDL was positively associated with coronary artery calcium (*p* < 0.01), and the authors suggest that the measurement of lipoprotein subclasses may improve the prediction of CAD in postmenopausal women beyond that provided by the conventional lipid panel and CAD risk factors [33].

## 5. Evidence for Treatment Response

Various lifestyle and pharmacologic treatments have been shown to have a differential effect on LDL subclass distribution. In general, treatments that tend to reduce fasting triglycerides tend to reduce the amount of small LDL. It was initially observed, by Dr. Peter Woods and colleagues at Stanford University, that individuals who chronically exercised by running had significantly less small LDL than matched individuals who did not exercise routinely [45]. This observation led to a series of clinical trials that determined that the expression of the small LDL trait was linked to percent body fat. In general, the greater the percent body fat, the greater the expression of small LDL and loss of excess body fat, through either exercise or caloric restriction, reduces small LDL significantly [46]. Concurrent investigations by Dr. Ronald Krauss and colleagues revealed that iso-caloric diets rich in simple carbohydrates induced expression of the small LDL trait and that the elimination of simple carbohydrates in the diet reduced small LDL [47]. Fish oil supplementation has been utilized to reduce elevated blood triglyceride levels. Fish oil supplementation has also been shown to reduce small LDL [48,49]. 

### Lipid Medications and LDL Heterogeneity

Medications such as beta-blockers that tend to increase fasting triglycerides also tend to increase small LDL [50]. Medications that reduce fasting triglycerides tend to reduce small LDL such as alpha blockers, niacin, and fibric acid derivatives [51,52,53]. The combination of a fibrate and niacin can be effective in small LDL reduction and avoid the need for high dose niacin [54]. These medications have a differential effect on the balance of small versus large LDL reduction. For example, in a study utilizing 3000 mg/d immediate release niacin per day, LDL-C was reduced 20%, but this masked a significant shift in LDL size distribution [52]. Small LDL was reduced 32%, counterbalanced by a 23% increase in large LDL, which resulted in little total LDL-C change but was a significant change in large versus small LDL. HMGCoA reductase inhibitor drugs can reduce small LDL, but generally the reduction is proportional to the total LDL-C reduction.

## 6. Evidence for Clinical Outcome Response

Investigations have reported a difference in clinical outcome in patients with an abundance of small LDL. These studies included the Stanford Coronary Risk Intervention Project (SCRIP), the Familial Atherosclerosis Treatment Study (FATS), the St. Thomas’ Atherosclerosis Regression Study (STARS), and the PLAC-I trial [31,55,56,57]. In SCRIP, a predominance of small LDL particles predicted coronary angiographic benefit from multi-risk factor intervention. In all these studies, therapeutic modulation of LDL size was associated with significantly reduced CHD risk on univariate analysis. Under multivariate analysis with adjustments for confounding factors, changes in LDL size by drug therapy were the best correlates of changes in coronary stenosis in FATS [55]. In STARS, the smallest LDL fraction was the plasma lipoprotein subfraction, with the single most powerful effect on coronary artery disease regression in middle-aged men with hypercholesterolemia [56]. The SCRIP study reported that despite almost identical LDL-C reduction in patients with predominantly dense (small) or buoyant (large) LDL particles, there was no significant arteriographic change difference between treatment and control patients with large LDL, whereas a significant reduction in the rate of arteriographic progression was seen in the treatment versus control in small LDL patients [31]. In PLAC-I using a logistic regression model that adjusted for lipid levels and other confounding factors, elevated levels of small LDL were associated with a nine-fold increased risk of coronary artery disease progression, but only in the placebo group [57]. In 2015, the JUPITER investigation revealed that dramatic reduction in LDL-C reduced the risk for a CHD event associated with small LDL, but the risk of a CHD event and mortality remained significantly associated with small LDL [37]. In patients with CHD and low HDL-C, the HATS trial reported that when adjusted for risk factors, the odds for primary clinical CHD endpoints were significantly greater in subjects with higher on-study small LDL levels both before (*p* = 0.01) and after (*p* = 0.03) adjustment for the treatment group and the standard lipid values [58]. Several laboratory methods are currently in use to assess LDL subclass distribution. An analysis of HATS samples by four different laboratory methods confirms the association of small LDL with greater CHD progression and confirms that the associations are independent of standard lipid measurements [14]. 

### Small Dense LDL and Interventional CV Outcomes

Small dense LDL-C may also predict outcome from interventional cardiovascular procedures. Kim and colleagues have reported, in a study of 412 drug eluting and bare metal stent patients, that in-stent restenosis (ISR) was significantly lower (*p* = 0.004) in patients with increased LDL particle size and LDL size was an independent predictor for ISR [58]. Miyazaki and colleagues have reported decreased in-stent intimal hyperplasia when fenofibrate inhibited cholesterol ester transfer activity and reduced the amount of small LDL [36]. Intravascular ultrasound (IVUS) can be used to visualize coronary plaque volume. In a 100 patient IVUS trial of single versus dual lipid lowering therapy, it was reported that lower total cholesterol, LDL-C, triglyceride, and stronger reduction of small dense LDL-C were observed in patients with plaque regression compared to those with progression [59]. 

## 7. Guidelines and Small, Dense LDL

In 2011, the National Lipid Association published advice from an expert panel that did not support the clinical utility of determining lipoprotein subclasses [60]. However, in 2011 it was also reported, from the European Consensus Statement on LDL subclasses, that several lines of evidence suggest that the quality of LDL influences cardiovascular risk [61]. To date, the magnitude and independence of the association of small, dense LDL with CHD has been tested in more than 50 studies, including cross-sectional and prospective epidemiologic as well as clinical intervention trials. Many, but not all of these trials demonstrate a significant association of small, dense LDL with increased cardiovascular risk or progression. Subsequent to the publication of these two guideline documents, several studies were published that strengthen the evidence that measurement of small, dense LDL has clinical utility and includes ARIC, MESA, JUPITER, Japan Secondary Prevention, HATS secondary prevention, and HATS 4 independent lab methods [14,25,26,37,42]. Most recently, a Japanese investigation of eight years in length confirmed that subjects in the highest quartile of sdLDL-C (≥43.7 mg/dL), had a 5.4-fold higher risk of CHD than those in the lowest quartile (≤24.4 mg/dL), and that sdLDL-C measurement significantly (*p* < 0.001) improves net reclassification [62]. Most recently, in 2020, investigators from the Copenhagen General Population Study reported, in 38,322 individuals, that subjects with higher sdLDLC (1 mmol/L, 39 mg/dL) had higher ASCVD risk [63].

## 8. Conclusions

Atherosclerosis is enhanced by many factors, one of which is an overabundance of small, dense LDL particles. The presence of these small, dense LDL particles is not apparent from standard LDL cholesterol laboratory methods. In the primary prevention population, an abundance of small LDL increases CHD risk 2–3-fold. An abundance of small, dense LDL is found in 30–40% of the CHD patient population. Elevation of sdLDL-C in stable CHD patients identifies a group at increased risk of a future CHD event. 50+ years of research has consistently revealed that this is a major risk factor for CHD events, and is often independent of traditional CHD risk factors. Expression of small LDL has a strong environmental component, and treatment is often the least expensive and includes reduction of excess body fat, avoidance of simple carbohydrates in the diet, exercise, niacin, fibric acid derivatives and omega-3 fish oil. Specific patient subgroups may benefit from sdLDL-C analysis. Studies have indicated that in patients with existing CHD, an abundance of small, dense LDL predicts disease progression, and reduced levels of small LDL portend a better cardiovascular outcome.

## Figures and Tables

**Figure 1 biomedicines-10-00829-f001:**
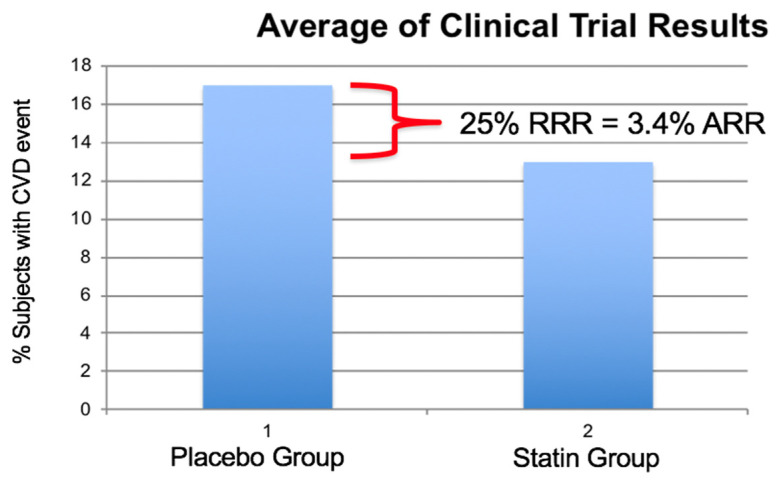
Percent of subjects experiencing a cardiovascular event in eight large statin investigations Scandinavian Simvastatin Survival Study (SSSS), Pravastatin or Atorvastatin Evaluation and Infection Therapy–Thrombolysis in Myocardial Infarction 22 (PROVE-IT 22), Heart Protection Study (HPS), Long-Term Intervention with Pravastatin in Ischaemic Disease (LIPID), Cholesterol and Recurrent Events (CARE), Treating to New Targets (TNT), Air Force/Texas Coronary Atherosclerosis Prevention Study (AFTEXCAPS), West of Scotland Coronary Prevention Study (WOSCOPS). Average relative risk reduction was 25% and the absolute risk reduction was 3.4% illustrating the large number of subjects experiencing a cardiovascular (CV) event while on statin therapy with reduced LDL-C values. (Modified from reference [1]).

**Figure 2 biomedicines-10-00829-f002:**
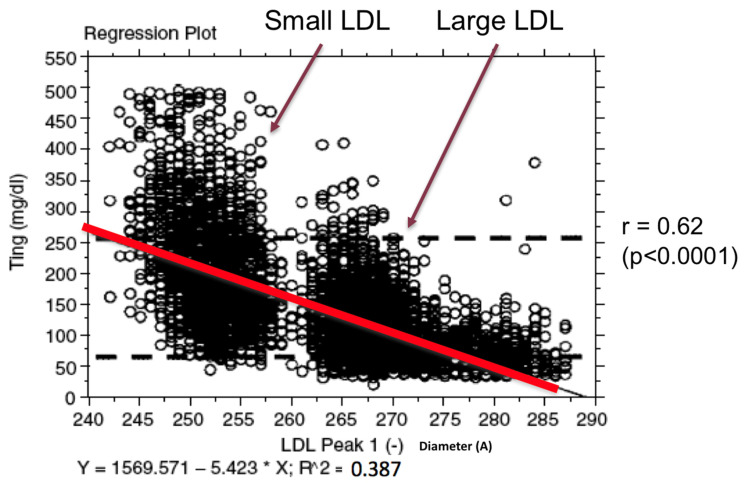
Scatter-plot of fasting triglycerides and LDL peak particle diameter in angstroms (*r* = 0.62, *p* < 0.0001) in 5366 CHD patients seen at the Fuqua Heart Center in Atlanta, Georgia. Large LDL particles have a diameter ≥ 263 angstroms and small LDL particles a diameter ≤ 257 angstroms.

**Figure 3 biomedicines-10-00829-f003:**
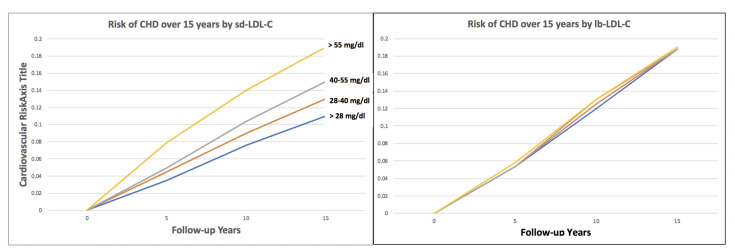
Small dense LDL-C (sd-LDL-C) quartiles and large buoyant LDL-C (lb-LDL-C) quartiles and cardiovascular risk over 15 years of follow-up in ARIC. (Modified from reference [25]).

**Table 1 biomedicines-10-00829-t001:** Relevant clinical investigations that have contributed to understanding the role of small, dense LDL in cardiovascular risk ranked by date of publication.

Study	Year	Findings
Framingham [9]	1966	Trig rich Sf20-400 lipoproteins associated with CAD risk
Lawrence Livermore [6]	1966	Trig rich Sf20-100 lipoproteins associated with CAD risk
NHLBI-II [28]	1987	IDL linked to arteriographic progression of CAD
Boston Area Heart [18]	1988	LDL pattern B associated with 3-fold increased CAD risk
STARS [29]	1992	Dense (small) LDL best predictor of arteriographic outcome
Physician’s Health survey [21]	1996	LDL pattern B associated with 3.4-fold increased CAD risk independent of total and HDL cholesterol and apo B
Stanford 5 City Project [20]	1996	LDL size best predictor of CAD risk by conditional logistic regression
MARS [30]	1996	In statin treated subjects with LDL-C < 85 mg/dL, triglyceride-rich lipoproteins were correlated with disease progression
SCRIP [31]	1996	Dense LDL predicts arteriographic benefit in the Stanford Coronary Risk Intervention project.
Quebec CV Study [19]	1997	Small LDL related to CHD risk.
		Statistical adjustment for LDL-C, triglycerides, HDL-C, and apoB had virtually no impact on the relationship of small LDL and CHD risk.
CARE [32]	2001	Large LDL size was an independent predictor of CHD events. Identifying patients on the basis of LDL size may not be useful clinically since pravastatin effectively treats risk associated with large LDL.
Healthy Women Study [33]	2002	Small low-density lipoprotein (LDL) was positively associated with coronary artery calcium (*p* < 0.01), but medium and large LDL were not.
SCRIP [34]	2003	Small low-density lipoprotein III but not low-density lipoprotein cholesterol is related to arteriographic progression
EAST [35]	2003	Arteriographic CAD progression over three years was significantly and independently linked to small, dense LDL particles.
Healthy Women Study [23]	2009	CVD risk prediction associated with lipoprotein profiles evaluated by NMR was comparable but not superior to that of standard lipids or apolipoproteins
HATS [36]	2014	Four laboratory methodologies confirm the association of small, dense LDL with greater coronary atherosclerosis progression and the associations were independent of standard lipid measurements.
ARIC [25]	2014	sdLDL-C was associated with future CHD events even in individuals considered at low CVD risk based on their LDL-C level.
MESA [26]	2014	sdLDL-C significantly associated with CHD risk even in subjects with LDL-C < 100 mg/dL who were normoglycemic
JUPITERr [37]	2015	Baseline LDL-C was not associated with CVD events, in contrast with significant associations for non-HDL-C and atherogenic particles including select subfractions of LDL particles.
AIM-HIGH [38]	2016	Levels of HDL3-C, but not HDL-C, HDL2-C, sdLDL, or LDL-TG, predict CV events in patients with metabolic dyslipidemia.
Malmo Heart [39]	2017	Smaller LDL particles are associated with incident CVD independently of traditional risk-factors, including standard lipids
Sakai [40]	2018	sdLDL-C was the most effective predictor of residual risk of future CHD events in stable older male CAD patients using statins and was independent of LDL-C
Copenhagen Heart Study [30]	2020	Individuals with high sdLDL-C had higher MI and ASCVD risk in 38,322 subjects.

STARS = Saint Thomas Arteriographic Regression Trial; HATS = HDL Atherosclerosis Treatment Study; ARIC = Atherosclerosis Risk in Communities; MESA = Multi Ethnic Study of Atherosclerosis; JUPITER = Justification for the Use of Statins in Prevention: An Intervention Trial Evaluating Rosuvastatin; EAST = Emory Angioplasty and Surgery Trial; SCRIP = Stanford Coronary Risk Intervention Project; AIM-HIGH = Atherothrombosis Intervention in Metabolic Syndrome with Low HDL/High Triglycerides and Impact on Global Health Outcomes (AIM-HIGH) trial.

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
