# Peer review of "Small Dense LDL: Scientific Background, Clinical Relevance, and Recent Evidence Still a Risk Even with ‘Normal’ LDL-C Levels"

_biomedicines, 2022, doi:10.3390/biomedicines10040829_

Round 1

Reviewer 1 Report

This is an interesting and informative review. The paper is generally well written, and I don’t have any major comments. The only thing that requires a minor revision is the use of abbreviations throughout the text.

For example:

in the abstract (line 10) LDL-C is not defined;

figure 1 legend (line 33) CV is not defined;  

small, dense LDL is used first in line 51, then abbreviated in line 63 (sdLDL), further in the text both the full term and the abbreviation are used randomly.

Line 74 - surfactant phospholipase assay (sdLDL-C) is this method specific for sdLDL-C? the usage in this form might be confusing since a line above the authors mention other methods with their abbreviations.

Nuclear magnetic resonance- first mentioned in lines 74-75, then abbreviated in line 86.

Those were the ones I noticed, but I would suggest the authors to check the consistency of all the abbreviations.

Since there are many abbreviations in the text, maybe it would be easier for a general reader to have a section with abbreviations listed, or perhaps some of them can be skipped by giving a full definition to facilitate the reading.

Author Response

Reviewer #1

Comments and Suggestions for Authors

* Author response.

This is an interesting and informative review. The paper is generally well written, and I don’t have any major comments. The only thing that requires a minor revision is the use of abbreviations throughout the text.

For example:

in the abstract (line 10) LDL-C is not defined;

*LDL-C name now printed in full.

figure 1 legend (line 33) CV is not defined;  

* CV has been changed to cardiovascular (CV)

small, dense LDL is used first in line 51, then abbreviated in line 63 (sdLDL), further in the text both the full term and the abbreviation are used randomly.

*small, dense LDL now has (sdLDL) inserted

Note: the use of sdLDL is intended to convey any lab method that assesses small LDL where as sdLDL-C is specific to measuring the cholesterol quantity on the small dense region.

Line 74 - surfactant phospholipase assay (sdLDL-C) is this method specific for sdLDL-C? the usage in this form might be confusing since a line above the authors mention other methods with their abbreviations.

* I agree the lab method abbreviations should be consistent to the method. sdLDL-C has now been changed to (SPA) to reflect the lab method of surfactant phospholipase assay.

Nuclear magnetic resonance- first mentioned in lines 74-75, then abbreviated in line 86.

* (NMR) was inserted after nuclear magnetic resonance, and, IM inserted after ion mobility.

*NMR was used in line 86

Those were the ones I noticed, but I would suggest the authors to check the consistency of all the abbreviations.

Since there are many abbreviations in the text, maybe it would be easier for a general reader to have a section with abbreviations listed, or perhaps some of them can be skipped by giving a full definition to facilitate the reading.

Reviewer 2 Report

The role of small dense LDL seem to be forgotten, although recently analyzed (Vekic J, Zeljkovic A, Cicero AFG, Janez A, Stoian AP, Sonmez A, Rizzo M. Medicina (Kaunas).  Atherosclerosis Development and Progression: The Role of Atherogenic Small, Dense LDL.; Jin X, Yang S, Lu J, Wu M. Small, Dense Low-Density Lipoprotein-Cholesterol and Atherosclerosis: Relationship and Therapeutic Strategies. Front Cardiovasc Med. 2022 Feb 10;8:804214. doi: 10.3389/fcvm.2021.804214. eCollection 2021). Authors in a clear way explain the significance of small dense LDL elevation, despite normal level of other lipoproteins. Historical trials presentation is another interesting part of the manuscript; however:

1) please try to use words in less casual, like 'heart attacks' instead of myocardial infarction/coronary syndrome (page 61); 'statin drugs' (page 279);

2) please do not repeat unnecessary abbreviations or explain them when used first time in the main text, i.e. CHD (line 44 and 63);

3) can Table 1 have frames? text seem to fuse;

4) there are some minor spelling errors. i.e. trigylcerides (line 111).

Author Response

Reviewer #2

Comments and Suggestions for Authors

* Author response.

The role of small dense LDL seem to be forgotten, although recently analyzed (Vekic J, Zeljkovic A, Cicero AFG, Janez A, Stoian AP, Sonmez A, Rizzo M. Medicina (Kaunas).  Atherosclerosis Development and Progression: The Role of Atherogenic Small, Dense LDL.; Jin X, Yang S, Lu J, Wu M. Small, Dense Low-Density Lipoprotein-Cholesterol and Atherosclerosis: Relationship and Therapeutic Strategies. Front Cardiovasc Med. 2022 Feb 10;8:804214. doi: 10.3389/fcvm.2021.804214. eCollection 2021). Authors in a clear way explain the significance of small dense LDL elevation, despite normal level of other lipoproteins. Historical trials presentation is another interesting part of the manuscript; however:

1) please try to use words in less casual, like 'heart attacks' instead of myocardial infarction/coronary syndrome (page 61); 'statin drugs' (page 279);

* “heart attacks” was replaced with ?myocardial infarction.

* line 284 “statin drugs” was replaced with “HMGCoA reductase inhibitor drugs”

2) please do not repeat unnecessary abbreviations or explain them when used first time in the main text, i.e. CHD (line 44 and 63);

* line 43 – “coronary heart disease (CHD)” was used in the abstract line 11

3) can Table 1 have frames? text seem to fuse;

* I am unable to determine how to place a frame on the image.

4) there are some minor spelling errors. i.e. trigylcerides (line 111).

 * Spelling was corrected and document spell checked again.

Reviewer 3 Report

The manuscript of Superko et al. aims to gives a comprehensive review of the clinical relevance of low-density lipoprotein heterogeneity and small dense LDL in the context of the effectiveness of treatment for cardiovascular outcomes. The topic is of current clinical interest and fits the scope of “Biomedicines”. Authors cite a comprehensive set of references with appropriate scientific coverage and including very recent literature up until 2021.

The manuscript is written in a very clear fashion and pleasant to read. It provides a historical overview of the studies establishing the connection between LDL cholesterol and atherosclerosis and cardiovascular outcomes as well as a balanced summary of evidence supporting the importance of small dense LDL fraction in these pathologies. The manuscript also cites studies with contradictory results and puts them in context in a relevant fashion.

There are some issues however, that need to be addressed before the manuscript could be considered for publication

  1. The title is extremely long and authors might reconsider rephrasing it in order to emphasise their core message about the importance of small dense LDL fraction in cardiovascular endpoints.
  2. The manuscript is illustrated by 3 figures, all referenced as “modified from other published work”. This reviewer had no access to Ref. 17 (Figure 2) so could not judge the degree of similarity. By contrast, Figure 3 is almost identical to the figure already published in Ref. 26.

Please check with the journal “Biomedicines” for potential copyright issues.

  1. Figure 1: It would be helpful to present the data in a boxplot format that would allow the visualisation of differences between the individual percentages obtained in each trial and not just the average without any data about the variability between trials.
  2. It would be extremely helpful to summarise the most important points/conclusions concerning the sdLDL fraction and CVD outcomes in a graphical abstract.

Author Response

Reviewer #3

Comments and Suggestions for Authors

* Author response.

The manuscript of Superko et al. aims to gives a comprehensive review of the clinical relevance of low-density lipoprotein heterogeneity and small dense LDL in the context of the effectiveness of treatment for cardiovascular outcomes. The topic is of current clinical interest and fits the scope of “Biomedicines”. Authors cite a comprehensive set of references with appropriate scientific coverage and including very recent literature up until 2021.

The manuscript is written in a very clear fashion and pleasant to read. It provides a historical overview of the studies establishing the connection between LDL cholesterol and atherosclerosis and cardiovascular outcomes as well as a balanced summary of evidence supporting the importance of small dense LDL fraction in these pathologies. The manuscript also cites studies with contradictory results and puts them in context in a relevant fashion.

There are some issues however, that need to be addressed before the manuscript could be considered for publication

  1. The title is extremely long and authors might reconsider rephrasing it in order to emphasise their core message about the importance of small dense LDL fraction in cardiovascular endpoints.

* I agree the title is a bit cumbersome and it has been shortened to “Small Dense LDL: Scientific Background, Clinical Relevance, and Recent Evidence Still a Risk Even with ‘Normal’ LDL-C Levels”

  1. The manuscript is illustrated by 3 figures, all referenced as “modified from other published work”. This reviewer had no access to Ref. 17 (Figure 2) so could not judge the degree of similarity. By contrast, Figure 3 is almost identical to the figure already published in Ref. 26.

*Figure 3 has been redrawn from the original figure 3 data

Please check with the journal “Biomedicines” for potential copyright issues.

  1. Figure 1: It would be helpful to present the data in a boxplot format that would allow the visualisation of differences between the individual percentages obtained in each trial and not just the average without any data about the variability between trials.

* The original Figure 1 was first published in the Circulation article cited and used multiple times in educational forums. We present the data as an average to avoid potential copyright conflicts. I would refer readers to the original figure in: Superko HR Circulation 2008;117:560-568.

  1. It would be extremely helpful to summarise the most important points/conclusions concerning the sdLDL fraction and CVD outcomes in a graphical abstract.

* It is unclear to me what the reviewer is referring to as a “graphical abstract”.  We tried to summarize the salient points in the Conclusion section.